# A Collaborative Approach to Understanding the Intersections of Practice and Policy for Peers in the Alcohol and Other Drugs Sector

**DOI:** 10.3390/ijerph21091152

**Published:** 2024-08-30

**Authors:** Timothy Piatkowski, Emma Kill

**Affiliations:** 1School of Applied Psychology, Mount Gravatt Campus, Griffith University, Brisbane, QLD 4122, Australia; 2Griffith Centre for Mental Health, Griffith University, Mount Gravatt, QLD 4122, Australia; emmak@quivaa.org.au; 3Queensland Injectors Voice for Advocacy and Action, Fortitude Valley BC, QLD 4006, Australia

**Keywords:** drugs, drug policy, lived–living experience, peers, steroids

## Abstract

Introduction: Peers in the alcohol and other drug sectors possess lived–living experience (LLE) crucial for shaping community care. However, genuine consumer collaboration is often confounded by stigma. This study examined peers’ perceptions, exploring their experiences regarding workforce dynamics, policy implications, and impacts on health equity. In presenting the research, we sought to synthesise the research methods and illustrate the methodological innovation and knowledge production in substance use research through authentic collaboration. Methods: We purposively sampled peer networks and community organisations, involving peer-researchers in planning, design, and analysis. We conducted semi-structured digital interviews with 18 peers and applied iterative coding to analyse the data. Results: This collaborative process provided nuanced insights into sectoral challenges. Peers expressed emotional strain revisiting personal substance use experiences, blurring personal and professional boundaries. Tokenistic peer involvement critiques underscored the need for genuine leadership and organisational support. Conclusion: We advocate for a shift towards equitable and inclusive policy development through both organisational and systemic restructuring. However, these changes are hamstrung by broader policy frameworks, which require a shift to peer-led principles, ensuring the expertise of peers is genuinely valued. Policymakers should invest in expanding peer frameworks, acknowledging the diversity within communities of people who use drugs to improve health equity and public health outcomes. This innovative approach to substance use research emphasises the transformative impact of integrating LLE into research.

## 1. Introduction

Community engagement has emerged as a pivotal strategy in shaping fair and inclusive health policies, practises, and research endeavours globally [1,2]. While community engagement encompasses diverse approaches, it often involves participatory elements in social planning, policy advocacy, and operational decision-making [3]. Central to this collaborative approach is the fundamental principle that people should have meaningful opportunities to contribute to decisions that directly impact their lives [3,4,5]. Peer engagement represents a distinct working context, where individuals are sought out for their firsthand experience (lived–living experience: LLE) with substance use. Introduced through harm reduction and ‘drug user movements’, peer engagement aims to elevate the voices of people who use drugs (PWUDs) in decision-making processes concerning them [6]. PWUDs’ expertise is leveraged to enhance the relevance and acceptability of various programmes, advocacy efforts, policymaking initiatives, and research endeavours across different domains [7]. Particularly in harm reduction contexts, peers have played integral roles in shaping and implementing programmes such as methadone maintenance, hepatitis C prevention, substance use treatment, and overdose prevention [8,9,10,11,12,13]. Today, PWUDs assume diverse roles in harm reduction work, including peer education, direct harm reduction service provision, group counselling facilitation, research support, and advisory committee participation [14,15,16]. Leveraging their LLE, peers offer valuable insights into the realities of drug use environments, ensuring that strategies and interventions remain relevant and acceptable [2,17]. These benefits must be viewed within the broader context of the stigmatisation and marginalisation faced by PWUDs. PWUDs often encounter, witness, and anticipate discrimination and mistreatment within professional healthcare and social services, leading to misunderstandings, distrust, and defensiveness from both parties even before they engage with these systems [17,18,19,20]. Nonetheless, the increased recognition of peer engagement as best practice in harm reduction affords communities more potential for the incorporation of the ‘peer lens’ into policy and programme frameworks. However, this expansion has also brought about challenges related to ensuring equitable representation, professional integration, and sustainable support for peers within broader health systems.

In contrast to self-help, mutual-aid, or peer-run organisations, which are primarily driven by peers themselves, many mainstream service settings not ‘led’ by peers may have limited prior exposure to or understanding of peer work. When employed in these emerging peer positions, people with LLE may face various challenges, including validating their experiential expertise within a professional-dominated environment, collaborating with non-peer colleagues, adapting to formal organisational protocols, and navigating conflicting expectations stemming from their dual personal and professional identities [17,21]. However, the integration of peers into non-peer-led mainstream organisations can also offer distinct advantages to both the organisations and the peers involved. Foregrounding the unique shared experiences of peers, recent work has demonstrated that incorporating peers into non-peer-led settings can yield benefits at the client, organisational, and societal levels while fostering peers’ personal and professional growth [17]. However, the multidimensional challenges arising from the unique peer identity, such as triggering, boundary negotiation, and feelings of entrapment, also warrant further attention [17]. The most recent recommendations have suggested effectively integrating peers into the current systems; however, organisations require collaboration with them to redefine organisational missions, cultures, and structures in ways that genuinely acknowledge and commit to the unique values of peers. We sought to extend this recent research in an applied setting.

This study sought to explore the evolving knowledge, skills, and experiences of peer workers in the alcohol and other drugs sector, particularly focusing on peer work strategies, wellbeing, reflective practice, boundaries, and disclosure. Additionally, we sought to assess the perceived impact of peer workers’ ongoing LLE on their own wellbeing, particularly in terms of mental health, and how these insights can inform policy and practice within the rapidly emerging alcohol and other drug (AOD) peer work sector.

## 2. Materials and Methods

### 2.1. Design and Ethics

This study is based on in-depth interviews. The research team comprised two peer-researchers who are on the board of a drug user organisation. These two peer-researchers implemented a collaborative approach, utilising their LLE to inform research aim development. This meant authentic peer involvement from the initial planning, defining the research design, conducting research, analysing data, and conveying findings [22,23]. Ethical clearance was obtained from the University Human Research Ethics Committee (2023/782).

### 2.2. Sampling and Recruitment

Participants (*N* = 18) were a community sample recruited via purposive sampling using the established peer networks of the investigators, which included peak alcohol and other drug organisations and service providers. Recruitment involved tapping into these professional networks, consulting with peers, and using social media and face-to-face interactions to promote the research. Potentially interested parties reached out to the research team or were contacted via email to arrange the interview at the convenience of the participant. Participants were included if they identified as peers with LLE of illicit drug use and had engagement working in the AOD sector. Prospective participants received a clear information sheet and could decline participation. Those who consented underwent individual interviews with recorded verbal consent. Participants were assured of their right to withdraw at any stage, and an AUD 50 gift card was offered as gratitude for their time and insights. The participants, with a mean age of 44.9 years (*SD* = 8.4), had an average of 9.5 years (*SD* = 8.4) of experience in formal peer work. The group included 15 females, 1 male, 1 trans female, and 1 non-binary person, with interview lengths averaging approximately 1 h.

### 2.3. Data Collection

All interviews were conducted by a single interviewer between November 2023 and February 2024 on Microsoft Teams, with no other researchers or nonparticipants present. The interview guide was developed by the research team and informed by both the extant literature and LLE. The interviews followed a semi-structured format, consisting of open-ended questions around peer workers’ role in the AOD sector. Example questions included the following: In your role as a peer within the community, how have your knowledge and skills evolved or improved, particularly in terms of peer work strategies, wellbeing, reflective practice, boundaries and disclosure? How do you perceive the impact of your involvement in the peer community on your own wellbeing, including aspects like mental health? To validate the interview questions, a pilot interview was undertaken with a member of the peer community. Its purpose was to confirm the questions’ alignment with the research goals and to refine the interview process. The pilot interview confirmed that the semi-structured interview guide was appropriately constructed. Interviews ranged between 45 and 80 min in length. The average length of the interviews was 65 min. Interviews were arranged according to participant availability and conducted via the Microsoft Teams platform, with automatic transcription. Participants were notified of the recording prior to the interview, and consent to participate was obtained both upon registration and verbally at the beginning of the interview. All interviews were conducted by one researcher, a female post-doctoral researcher with post-graduate training in qualitative research. Transcripts were then manually reviewed by the two authors to ensure accuracy before being imported into NVivo for analysis.

### 2.4. Data Analysis

Data analysis was conducted using iterative categorisation, as per Neale [24], in NVivo (v12, QSR) qualitative analysis software. Data were collected, coded, and analysed using an iterative process, where recruitment ceased when the data no longer added anything new to the overall analysis. Transcripts were offered to participants for review or comment; however, no participants requested this. Initially, deductive codes were formulated based on the predetermined topics of interest outlined in the interview guide, such as ‘peer work challenges’, ‘lived–living experience’, and ‘boundaries’. Additionally, inductive codes, including ‘diversity’, ‘capacity’, ‘capability’, and ‘mental health’, were generated through ongoing team discussions addressing developed theme-categories. The research team did not seek to establish interpretive consensus [25]; however, they held various meetings to discuss the way in which theme-categories were understood and applied. Following this, the first author applied the finalised codes to all interview transcripts. Analysis focused on reviewing codes related to specific topics and peer workforce challenges, identifying the emergent analytic themes which had been documented [26]. 

The next section presents the findings of these analyses. We focus on three overarching theme-categories: Re-framing the peer work landscape; re-aligning peer representation, top-down policies, and workforce development; and considerations surrounding peer diversity.

## 3. Results

### 3.1. ‘Most of Us Don’t Stop Being Lived Experience Experts’: Re-Framing the Peer Work Landscape

Peer workers in the AOD sector face persistent challenges, as illustrated by our participants’ accounts. Peers provided accounts of the emotional labour involved in peer work, particularly in continually revisiting personal experiences related to service use and interactions. 


*Ela [60, female]: Really exhausting. Hard mental lifting. To keep dragging up your experiences of using services or you know just doing some work with the police, dragging up your experiences of the police. I think that takes its toll as a peer. You have to keep that door open to your past and I think it really does take a big emotional toll.*


This perpetual openness can be emotionally taxing, as it requires peers to confront and process their own past traumas and challenges while simultaneously providing support to others. As a result, peers’ personal boundaries become blurred, leading to a perpetual state of being ‘on’ even in their personal lives. 


*Morgan [28, trans female]: I have definitely found being a peer worker that sometimes having my own boundaries has been quite difficult when like my next door neighbour has an extended mental breakdown or my neighbour a couple doors down has like her kids being arrested for stealing a car. And so like we all bundle into my apartment and I give them advice for their fucking meth [methamphetamine] breakdown and upcoming court dates. And that can be a bit draining at times and I think it’s part of the lived experience work stuff that a lot of people don’t recognise is that most of us don’t stop being lived experience experts.*


The lack of separation between personal and professional life can be emotionally laborious and may jeopardise peer workforce wellbeing. Moreover, what these perspectives underscore is the often-unacknowledged reality that people in peer roles are continually relied upon as LLE experts [1,17], perpetuating the expectation that they are always available to provide support and guidance. This phenomenon raises critical questions about the sustainability of peer work models and the necessity for clearer boundaries and support mechanisms to safeguard the health and resilience of peer workers.

Other peers also spoke of the dual nature of peer interactions within the AOD sector, a phenomenon documented in extant work [27]. 


*Olivia [44, female]: I had a relapse into, into ice [methamphetamine] use through another peer and that was really horrible. I think that there is that danger zone of, depending on where someone’s at, you can kind of drag each other down.*


Peers, while providing invaluable support, also face the risk of being influenced by others. However, the ‘double-edged’ [28] effect underscores the importance of comprehensive policies and supportive workforce environments that prioritise the safety and wellbeing of peer workers. Indeed, the complexity of boundary-setting in peer work was further highlighted. 


*Skyler [41, non-binary]: There’s stuff that like, my own personal boundaries, right? Like I wouldn’t use drugs with a client. That’s fair enough to say, right? But like what happens if you meet you a client at the dealer’s house? There’s all these kind of boundaries that you would never think about in another role.*


There are unique intricacies to navigating personal and ‘work’ boundaries, especially in the ‘risk laden’ [29] environments that exist for PWUDs. Translating these to a ‘workplace’ setting, certain boundaries may seem clear, such as refraining from using drugs with clients; however, the dynamic nature of peer work in the AOD sector introduces unforeseen challenges. For instance, encountering clients in settings like a ‘dealer’s’ house blurs the lines between professional ethics. These types of events require further consideration as they underscore the need for more nuanced approaches to boundary-setting in peer work. From both personal and workforce perspectives, it is crucial to acknowledge and address the myriad factors that influence boundary establishment and maintenance. Building on this, however, some peers highlighted the evolution of boundaries as a natural process in their roles. 


*George [48, male]: It’s harmful to not have boundaries, you know, and it’s [not] just for yourself but for the person. It just evolves kind of like, naturally I guess. Boundaries, around lots of stuff, disclosure, purposeful sharing, you know.*


This ‘on the job’ learning suggests that boundary-setting is a dynamic and evolving aspect of peer work influenced by personal experiences and professional development. Peers emphasised the importance of establishing boundaries not only for their own wellbeing but also for the benefit of the individuals they support. 

But what is occurring at a workforce and systemic level to keep peers safe and supported? We begin to trace these connections in the next section.

### 3.2. ‘Like a Box to Tick’: Re-Aligning Peer Representation, Top-Down Policies, and Workforce Development

Peers navigate personal challenges outside of work, but they also encounter professional hurdles which intersect at the systems level and are relevant to both the workforce and policies. Some of the current peer cohort expressed sentiments such as being undervalued and utilised for tokenistic purposes by organisations. 


*George [48, male]: You look at what they’re [organisations] actually doing, and it’s often they’re not even consulting with lived experience. A few things I’ve been involved in and they’ll claim that [lived experience consultation]. And then we’re [peers] consulted right at the end of it and not at the beginning. So, they kind of just want to, like, use us like a box to tick, you know?*


Being relegated to a mere ‘tickbox’ in decision-making processes undermines the genuine contributions of peer workers, diminishing their sense of worth and agency within the sector. Such tokenism perpetuates a cycle of marginalisation and disregards the valuable insights and experiences that peers bring to an organisation. 


*Skyler [44, female]: [Organisation name], you know, has done some really terrible things with peer workers and promised them things got them to jump through a whole heap of hoops and get police, clearance and having to unravel and share all of their past legal stuff, which is really traumatic. And then just dropping them, you know, and creating a world of pain for that person and just setting them up to fail like an absolute set up and that’s where I think the danger is.*


Peers spoke of the critical importance of genuine peer consultation and leadership within organisations. They did so by drawing attention beyond a workforce perspective, demonstrating a link to broader systems and environments, such as a policy context. 


*Rihanna [37, female]: Like, there is no reason that anyone should be in a position of managing people or managing people with lived experience if they’ve got no experience. Like, people who come into these organisations and they say things like, yeah, they’re real allies and yet, the practice is not correct. If we’re gonna be peer led, then we should be peer led.*


There appeared to be a perceived disconnect between organisational rhetoric and the actual practice, which fell short of genuine peer involvement and leadership. This disconnect could be indicative of a need for alignment between stated organisational values and operational practises. More importantly, it points to the necessity of emergent policies that prioritise peer-led approaches, recognising the unique expertise and perspectives that individuals with lived experience bring to the table. As such, it may be time to call for a paradigm shift towards genuine peer leadership across policy decision-making within the AOD sector. 

To provide some demonstrable elements for readers, we use the ‘*worked example*’ of the peer workforce in the AOD sector and the inherently risky socio-political environment they navigate. 

Some peers reflected around the intersection of workplace dynamics and broader policy frameworks within the AOD sector. They highlighted the pervasive nature of drug-related discussions and harm reduction efforts that permeate their professional interactions, regardless of their specific role. 


*Ela [60 female]: I do face to face work. I mean, there’s always people who use drugs and they always seem to find me whatever role I’m in, they end up talking drugs, harm reduction or whatever. But outside of that, I think most of my peer work has been very like systemic sort of activism in this area more than anything… within AOD, it’s the whole world you’re taking on. It’s a predominant narrative. It’s the newspapers, it’s the police, it’s the courts, it’s everybody.*


This suggests that the challenges and responsibilities faced by peers extend beyond individual interactions with clients to encompass systemic advocacy and activism within the AOD domain. Moreover, these data underscore the omnipresence of a narrative surrounding people who use drugs within various spheres, including media, law enforcement, and judicial systems, documented among previous LLE work [30,31]. These pervasive narratives shape not only public discourse but also institutional responses to drug-related issues, influencing policies and practises within the AOD sector. 

The discourse draws attention to the multifaceted nature of peer work, which involves not only direct client engagement, but also broader advocacy efforts aimed at addressing systemic barriers and promoting harm reduction initiatives. Participants further spoke about the intricate interplay between individual experiences and broader structural forces, emphasising the importance of approaching and addressing drug-related challenges beyond a workplace and more so within a policy context [32].


*Morgan [28, trans female]: Peer work has to be inherently political, and if you’re not doing like Peer work politically, you’re just not really doing peer work. And if you’re not coming from a space of anti-oppression—what the fuck are you doing?*


By framing peer work as inherently political, Morgan underscored the interconnectedness between individual support and broader advocacy for systemic change [33,34,35,36,37]. This type of assertion suggests that effective peer work extends beyond individual interactions with clients to include active engagement with political and policy domains. Moreover, the discourse further underscores the intersectionality of drug-related issues with broader systems of oppression and marginalisation. However, these discourses are not homogeneous or static, meaning that the influence of peer workers is dynamic and context-dependent, continually interacting with and reshaping socio-political environments.

### 3.3. ‘Specific Peer Spaces’: Considerations Surrounding Peer Diversity

The cohort underscored the necessity for diversity and specificity within peer spaces.


*Skyler [41, non-binary]: I thought, you know any drug, you know, as long as it’s a peer, it’s cool, right? But I think we need to have a whole like army of peers where everyone’s represented there.*


Participants observed misconceptions that any peer, regardless of their background or experience with drugs, is suitable for peer work. However, some peers extended this notion further, emphasising the importance of tailored peer spaces that cater to individuals with specific lived experiences, such as people who inject drugs. 


*Phoebe [50, female]: I’m an injecting drug user, so there’s a space there that I can be of most use in. I do think that there needs to be specific peer spaces.*


Together, these perspectives emphasise the need for a nuanced understanding of peer diversity and the recognition that a ‘one size fits all’ approach to peer inclusion is inadequate [38,39]. By acknowledging and valuing the unique contributions of peers with diverse backgrounds and experiences, organisations can create more inclusive and effective peer support networks that better meet the diverse needs of individuals within the AOD sector.

With some critical framing, we direct readers to the significance of addressing peer diversity issues. We saw this exemplified by the underrepresentation of certain peer groups in the current data, notably ‘[anabolic–androgenic] steroid peers’. The absence of steroid peers, underscored by participants, reflects a broader issue of overlooking these diverse perspectives within the AOD sector [40,41,42]. This serves as a pertinent example of the need to recognise and integrate various peer experiences to ensure inclusivity and effectiveness in peer support networks. The acknowledgment of the difficulty in providing adequate support to individuals within the steroid community highlights the systemic inadequacy in current support structures.


*Jacinta [34, female]: Especially with the steroid community as well. So it’s a really hard group like [first author’s] amazing at doing his work in that space. But I think, yeah, I definitely think there needs to be like lots of [first author]’s to be able to support people. ‘Cause, that’s not my strength.*


The call for multiple peers with specialised knowledge reflects not only the complexities of some types of drug consumption but also the failure of existing systems to accommodate diverse needs adequately. Other participants expressed personal limitations in engaging with particular drug consumption issues, thus exposing a critical gap in peer support networks. 


*Caitlin [48, female]: But I haven’t and the other peers at work they have not experienced steroid use so… we can read about it till the cows come home. But we’ve never done it. So, and we would love to get somebody on board who was a peer in that sphere… that would be amazing.*


Some participants expressed that desire to acquire firsthand experience in steroid use reveals the inadequacy of existing knowledge frameworks in addressing emerging drug trends. 


*Bonnie [45, female]: There’s only so much information I know about steroids and we have so many people coming and using steroids. It’ll be friggin awesome like I can just say like I would love to probably start using steroids so I can so I know a bit more first hand, you know what I mean? So you know, because we’ve got so many young people coming in all the time and I don’t know what to tell them. OK, well when I use them like it’s completely different to someone who injects you know use IV [intravenous] then intramuscular.*


These narratives underscore the importance of LLE, especially in the context of drug use. By expressing a desire to personally use steroids to gain firsthand insight, participants highlight the profound impact of experiential knowledge in enhancing understanding and empathy. The recognition of diverse drug consumption methods—intravenous, intramuscular, and oral administration, to name a few—emphasises the nuanced understanding derived from direct experiences. Peer workers, through their LLE, voices, and insights, can be mobilised to constantly challenge and reshape the socio-political contexts influencing drug-related issues.

## 4. Discussion

The nature of our analyses affords us a more comprehensive view of the evolving landscape of peer work within the AOD sector, highlighting both the challenges and opportunities faced by peer workers. These insights provide valuable implications for harm reduction, health policy, and workforce development. The broader practice of peer work encompasses individuals’ contributions to research, advocacy, policy development, and peer support. However, within policy and organisational representations, the role of peer workers is increasingly being confined to peer support alone, exemplified by a growing conflation of terms. Effective policy influence by peers, particularly from ‘drug user organisations’, depends on more than just the quality of their advocacy arguments; it also requires trust and alignment with non-peer organisations and policy networks [43]. These organisations, which often face multiple barriers, including stigma and limited resources [17,44], must navigate accountability and credibility issues within both their communities and the policy system [45,46]. Achieving meaningful impact involves aligning policy and service systems to support peer-based initiatives. We attempt to build on this previous work with the addition of our findings to the current discourse.

Firstly, the findings underscore the increased biopsychosocial ‘toll’ which comes from peer work. Peer workers often find themselves continually revisiting personal experiences related to substance use, which can be emotionally taxing [47]. This perpetual openness blurs the boundaries between personal and professional life, leading to challenges in maintaining self-care and wellbeing [27]. We believe this emphasises the need for clearer boundaries and support mechanisms to safeguard the health and resilience of peer workers. Organisations must prioritise the development of workforce policies and practises that promote self-care and provide adequate support for peer workers to prevent burnout and compassion fatigue. The findings underscore the need for diversity and specificity within peer spaces. A one-size-fits-all approach to peer inclusion is inadequate, as different peer groups may have unique needs and experiences. Organisations must recognise and value the diverse perspectives and contributions of peers from various backgrounds. This requires tailored peer support networks that cater to the specific needs of different peer groups, ensuring inclusivity and effectiveness in peer support initiatives.

### 4.1. Implications for Practice and Policy

These data highlight the importance of authentic peer leadership within organisations. Tokenistic approaches to peer involvement undermine the genuine contributions of peer workers and perpetuate feelings of marginalisation [1,2]. There is a clear need for organisational policies that prioritise genuine peer involvement and leadership, ensuring that peers have a meaningful voice in decision-making processes. However, we suggest that this requires a paradigm shift towards more equitable and inclusive approaches to policy development and organisational culture within the AOD sector—what we term here as a ‘top-down and peer-led approach’. However, to ensure clarity and consistency in implementation, this shift necessitates a broader restructuring of health and drug policy frameworks, removing decision-making from the sole purview of organisations. By embedding peer-led principles into overarching policy frameworks, we can foster a more cohesive and equitable approach to addressing substance use and harm reduction initiatives, thus ensuring that the voices and expertise of peer workers are genuinely valued and integrated into decision-making processes at all levels.

Peers with firsthand experience of emerging drug trends can provide invaluable insights and support to individuals within these communities. Organisations must prioritise the recruitment and training of peers with diverse experiences, including acknowledging those with firsthand experience of anabolic–androgenic steroid use [48]. This requires ongoing education and professional development opportunities for peer workers to enhance their knowledge and skills in supporting individuals with diverse drug use experiences. Moreover, it is crucial to recognise that peer work is a diverse field and requires ongoing adaptation and specialisation. Each substance and pattern of use presents its own set of challenges and complexities, requiring tailored approaches to the support strategies offered. By investing in recruitment of diverse peers, as well as ongoing education and professional development opportunities for current peer workers, organisations can ensure that they are equipped with a workforce which has the knowledge, skills, and cultural competence necessary to effectively engage with people across a wide spectrum of substance use experiences.

### 4.2. Limitations

While this study aimed to capture the diverse experiences of peer workers in the AOD sector, it is essential to acknowledge the potential limitations inherent in our sampling strategy. The use of purposive sampling may have introduced selection bias, as participants were recruited primarily through established peer networks and may not fully represent the broader population of peer workers in the AOD sector. While our findings offer rich insights into the experiences of peer workers, caution should be exercised in extrapolating these findings to other settings or populations outside of Australia. 

## 5. Conclusions

This research has illuminated the multifaceted landscape of peer work within the AOD sector, revealing both the challenges and opportunities faced by the LLE workforce. The findings underscore the emotional toll of peer work and the critical importance of establishing clear boundaries and robust support mechanisms to safeguard the wellbeing of peer workers. Moreover, the need for diversity and specificity within peer spaces highlights the need for innovative approaches toward peer inclusion, emphasising the importance of peer support networks that cater to the unique needs of different groups. Importantly, this study underscores the imperative for authentic peer leadership within organisations, advocating for a paradigm shift towards more equitable and inclusive approaches to policy development and organisational culture within the AOD sector. We propose the adoption of genuine peer involvement and leadership in decision-making processes. However, to ensure clarity and consistency in implementation, this shift necessitates a broader restructuring of health and drug policy frameworks, removing decision-making from the sole purview of organisations. Looking ahead, it is essential for organisations to prioritise the recruitment and training of peers with diverse experiences, including those with LLE of emerging drug trends such as anabolic-androgenic steroid use. By embedding peer-led principles into overarching policy frameworks and fostering ongoing education and professional development opportunities for peer workers, we can ensure that the voices and expertise of peer workers are genuinely valued and integrated into decision-making processes at all levels. We call for a concerted effort to elevate the role of peer workers within the AOD sector, recognising their invaluable contributions and advocating for policies and practises that support their wellbeing and professional growth. 

## Data Availability

Data are available from the authors upon reasonable request.

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
