# Peer review of "A Collaborative Approach to Understanding the Intersections of Practice and Policy for Peers in the Alcohol and Other Drugs Sector"

_ijerph, 2024, doi:10.3390/ijerph21091152_

Round 1

Reviewer 1 Report

Comments and Suggestions for Authors

The manuscript covers an important topic in the drug and alcohol field and the peer worker movement more broadly. It extends on growing work around the role and contribution of professional peer work in the grounded example of the drug and alcohol sector through interviews with peer workers. Mapping the competing demands, blurred lines and complex dynamics of peer work in the drug and alcohol sector is an essential contribtion to the literature. The authors do this with skill, and are to be commended for generating such rich, situated data about peer work. The impact of this material is embodies quotes like this: “if you're not coming from a space of anti-oppression. What the fuck are you doing?” - powerful stuff that really illustrates the issue being discussed in the paper. 

The above is a clear indication that the paper makes the required controibution to be considered for publication. With that in mind, there are elements of the manuscript that could be improved, which I outline below. 

First, it would be useful to mention that this is an interview study in the first section of the methods. It is mentioned later, but it is worth doing upfront.

Second, it is my view that the use of the ‘what is the problem represented to be?’ (WPR) analytical framework adds little to the overall contribution of the work, and may even detract from the key issue at the centre of the analysis (peer voices). The WPR framework is a method of analysing policy documents or positions - it steps researchers through the analysis of how policies come about, what they say (and what they assume), what they do in practice and so on. There is also the issue of the kind of analysis that WPR is asking for, which is to illustrate the often unspoken structural elements or power relations that rest behind the construction of policy problems. This calls for an analysis which illustrates how policy problems are made by the policy itself, rather than simply addressing pre-existing, self-evident problems. All of this is to say that this manuscript is not doing policy analysis, it is doing an analysis of interview data - and Bacchi has herself said that WPR is not well suited for the analysis of interview data (Bacchi & Goodwin, 2016).

Some examples of where the use of WPR to analyse the interview data is an issue may be useful. In the first data analysis section the authors say “it becomes evident that the representation of peer work often neglects the inherent complexities faced by Peers in this workforce” - while the data presented richly illustrates the complexity of peer work, the author does not provide a policy against which the reader can assess how peer work is represented, and therefore no way to understand if that representation is accurate (or not) to peer experiences. In another example, the authors say “Applying Bacchi's approach, we can reframe peer workers not just as supporters” - yet no not mention is made of what the ‘origional’ frame is that it would be useful to use WPR to reframe.

I am of the view that the WPR framework is not needed and could be removed. The analysis is of peer workers’ narration of their own experiences and these are valuable in and of themselves. There is no explicit need to analyse these against anything else, or to introduce a conceptual framework (like WPR) from which to undertake an analysis of what these narratives say about the state of peer work in the AOD field. To me, removal of this material is the most straightforward way to address the issue and would make the manuscript more impactful (and publishable). 

If the authors wish to keep the WPR framework in the analysis, then they would need to directly address the issues raised above in the method section and say how they are adapting WPR (a method for policy document analysis) to do an analysis of interview data. One option might be to conduct a short analysis of policy/policies around peer workers in the sector and to juxtapose this against the interview data - showing how the policy constructs the ‘problem’ against how peer workers experience it.

Author Response

Reviewer 1

The manuscript covers an important topic in the drug and alcohol field and the peer worker movement more broadly. It extends on growing work around the role and contribution of professional peer work in the grounded example of the drug and alcohol sector through interviews with peer workers. Mapping the competing demands, blurred lines and complex dynamics of peer work in the drug and alcohol sector is an essential contribtion to the literature. The authors do this with skill, and are to be commended for generating such rich, situated data about peer work. The impact of this material is embodies quotes like this: “if you're not coming from a space of anti-oppression. What the fuck are you doing?” - powerful stuff that really illustrates the issue being discussed in the paper. 

The above is a clear indication that the paper makes the required controibution to be considered for publication. With that in mind, there are elements of the manuscript that could be improved, which I outline below. 

Response: We thank the Reviewer for their time and consideration of our manuscript. We appreciate their enthusiasm for our work and the constructive feedback they have provided regarding the theoretical framing. We have attempted to revise the manuscript with their concerns in mind. Changes are outlined below and denoted in the manuscript in red font.

First, it would be useful to mention that this is an interview study in the first section of the methods. It is mentioned later, but it is worth doing upfront.

Response: Thank you, we have ensured to do this from the outset of the Methods.

Text added: This study is based on in-depth interviews. The research team comprised two peer-researchers who are on the Board of a Drug-User Organisation. These two peer-researchers implemented a collaborative approach, utilising their LLE to inform research aim development. This meant authentic peer involvement from the initial planning, defining the research design, conducting research, analysing data, and conveying findings
[28,29]. Ethical clearance was obtained from the University Human Research Ethics Committee (2023/782).

Second, it is my view that the use of the ‘what is the problem represented to be?’ (WPR) analytical framework adds little to the overall contribution of the work, and may even detract from the key issue at the centre of the analysis (peer voices). The WPR framework is a method of analysing policy documents or positions - it steps researchers through the analysis of how policies come about, what they say (and what they assume), what they do in practice and so on. There is also the issue of the kind of analysis that WPR is asking for, which is to illustrate the often unspoken structural elements or power relations that rest behind the construction of policy problems. This calls for an analysis which illustrates how policy problems are made by the policy itself, rather than simply addressing pre-existing, self-evident problems. All of this is to say that this manuscript is not doing policy analysis, it is doing an analysis of interview data - and Bacchi has herself said that WPR is not well suited for the analysis of interview data (Bacchi & Goodwin, 2016).

Some examples of where the use of WPR to analyse the interview data is an issue may be useful. In the first data analysis section the authors say “it becomes evident that the representation of peer work often neglects the inherent complexities faced by Peers in this workforce” - while the data presented richly illustrates the complexity of peer work, the author does not provide a policy against which the reader can assess how peer work is represented, and therefore no way to understand if that representation is accurate (or not) to peer experiences. In another example, the authors say “Applying Bacchi's approach, we can reframe peer workers not just as supporters” - yet no not mention is made of what the ‘origional’ frame is that it would be useful to use WPR to reframe.

I am of the view that the WPR framework is not needed and could be removed. The analysis is of peer workers’ narration of their own experiences and these are valuable in and of themselves. There is no explicit need to analyse these against anything else, or to introduce a conceptual framework (like WPR) from which to undertake an analysis of what these narratives say about the state of peer work in the AOD field. To me, removal of this material is the most straightforward way to address the issue and would make the manuscript more impactful (and publishable). 

If the authors wish to keep the WPR framework in the analysis, then they would need to directly address the issues raised above in the method section and say how they are adapting WPR (a method for policy document analysis) to do an analysis of interview data. One option might be to conduct a short analysis of policy/policies around peer workers in the sector and to juxtapose this against the interview data - showing how the policy constructs the ‘problem’ against how peer workers experience it.

Response: We thank the Reviewer for this comprehensive and considered approach to providing feedback. We appreciate their suggestion and have removed the WPR Approach from all components of the manuscript. This means, as the Reviewer has indicated, that the Peer voices and narratives are elevated and ‘front-and-centre’ of the paper. Please see Introduction, Findings, and Discussion to see changes in red font.

We thank the Reviewer not only for providing constructive feedback, but for suggesting multiple options regarding how best to approach the feedback.

Reviewer 2 Report

Comments and Suggestions for Authors

This paper explores the experiences of peer workers in the alcohol and other drugs sector, and the perceived impact of peer workers' ongoing LLE on their own wellbeing within the emerging area of AOD peer work. The aim of contributing knowledge to build approaches to policy development that empower  and value peer workers within the AOD sector. It provides useful new insights to this emerging area.

Intro

A good overview of the complexity of the area and the work of people who use drugs peer and LLE work within mainstream organisations, as well as a useful overview of the policy frameworks that guided the study, and the value given to different experiences and knowledge within policy processes.

A suggestion is perhaps more direct recognition in the introduction of the past work looking at the role of stigma impacting on the policy voice or influence of PWUD peer leadership. This is noted in the abstract but only inferred in the introduction/background.

Methods

Clearly described and appropriate methods.

Question – how many participants were involved – this seems to have been omitted, and any overview of the diversity of participants? Apologies if I have missed this.

Results

Good analysis with linking/alignment with the analytical frameworks, and clear themes described. While some of the findings are consistent with past work, a useful emphasis is made on the role of peer work not only direct client engagement, but also broader advocacy regarding systemic barriers and promoting harm reduction initiatives, and the insight of peer work being inherently political.

Discussion

The paper takes a useful application of Bacchi’s analytical frame, and provides useful and direct implications for practice.

Suggestion: Past work in this area in Australia and elsewhere has mainly focused on peer leadership from within peer led organisations. It may be useful to reference some of this work, highlighting how this study goes beyond this past work, looking at peer work in mainstream services, and provides new  perspective provided through the use of Bacchie framing in the analysis.

Author Response

Reviewer 2

This paper explores the experiences of peer workers in the alcohol and other drugs sector, and the perceived impact of peer workers' ongoing LLE on their own wellbeing within the emerging area of AOD peer work. The aim of contributing knowledge to build approaches to policy development that empower  and value peer workers within the AOD sector. It provides useful new insights to this emerging area.

Response: We thank the Reviewer for their time and thoughtful consideration of our manuscript. We appreciate their enthusiasm for our work and the constructive feedback. We have revised the manuscript to address their concerns, and the changes are outlined below and highlighted in red font within the manuscript.

Intro

A good overview of the complexity of the area and the work of people who use drugs peer and LLE work within mainstream organisations, as well as a useful overview of the policy frameworks that guided the study, and the value given to different experiences and knowledge within policy processes.

A suggestion is perhaps more direct recognition in the introduction of the past work looking at the role of stigma impacting on the policy voice or influence of PWUD peer leadership. This is noted in the abstract but only inferred in the introduction/background.

Response: Thank you for this suggestion, we have included more recognition of the literature which has looked at the role of stigma among PWUD. This has been added into the Introduction.

Text added:
These benefits must be viewed within the broader context of the stigmatisation and marginalisation faced by PWUDs. PWUD often encounter, witness, and anticipate discrimination and mistreatment within professional healthcare and social services, leading to misunderstandings, distrust, and defensiveness from both parties even before they engage with these systems [16-19]. Nonetheless, the increased recognition of peer engagement as best practice in harm reduction affords communities more potential for the incorporation of the ‘peer lens’ into policy and program frameworks.

Methods

Clearly described and appropriate methods.

Question – how many participants were involved – this seems to have been omitted, and any overview of the diversity of participants? Apologies if I have missed this.

Response: Apologies for this error and omitting this information. We have now included information regarding the participants in the Methods.

Text added:
Participants (N = 18) were a community sample recruited via purposive sampling using the established peer networks of the investigators, which included peak alcohol and other drug organisations and service providers. Recruitment involved tapping into these professional networks, consulting with Peers, and using social media and face-to-face interactions to promote the research. Potentially interested parties reached out to the research team or were contacted via email to arrange the interview at the convenience of the participant. Participants were included if they identified as Peers with LLE of illicit drug use and had engagement working in the AOD sector. Prospective participants received a clear information sheet and could decline participation. Those who consented underwent individual interviews with recorded verbal consent. Participants were assured of their right to withdraw at any stage, and a $50 gift card was offered as gratitude for their time and insights. The participants, with a mean age of 44.9 years (SD = 8.4), had an average of 9.5 years (SD = 8.4) of experience in peer work. The group included 15 females, 1 male, 1 trans female, and 1 non-binary person, with interview lengths averaging approximately 1 hour.

Results

Good analysis with linking/alignment with the analytical frameworks, and clear themes described. While some of the findings are consistent with past work, a useful emphasis is made on the role of peer work not only direct client engagement, but also broader advocacy regarding systemic barriers and promoting harm reduction initiatives, and the insight of peer work being inherently political.

Response: We thank the Reviewer for this feedback.

Discussion

The paper takes a useful application of Bacchi’s analytical frame, and provides useful and direct implications for practice.

Suggestion: Past work in this area in Australia and elsewhere has mainly focused on peer leadership from within peer led organisations. It may be useful to reference some of this work, highlighting how this study goes beyond this past work, looking at peer work in mainstream services, and provides new  perspective provided through the use of Bacchie framing in the analysis.

Response: We acknowledge the point made by Reviewer 2. However, in light of Reviewer 1’s constructive feedback, we have chosen to remove Bacchi’s analytical frame from this work.

We have, however, attempted to ensure a variety of citations from past Australian work focused on peer leadership and in peer-led organisations in the Discussion – thank you for this suggestion and directing us to this important work.

Text added: Effective policy influence by Peers, particularly from ‘drug user organisations’ depends on more than just the quality of their advocacy arguments; it also requires trust and alignment with non-peer organisations and policy networks [42]. These organisations, which often face multiple barriers including stigma and limited resources [16,43], must navigate accountability and credibility issues within both their communities and the policy system [44,45]. Achieving meaningful impact involves aligning policy and service systems to support peer-based initiatives. We attempt to build on this previous work with the addition of our findings to the current discourse.

Citations added:
Brown, G.; Perry, G.-E.; Byrne, J.; Crawford, S.; Henderson, C.; Madden, A.; Lobo, R.; Reeders, D. Characterising the policy influence of peer-based drug user organisations in the context of hepatitis C elimination. International Journal of Drug Policy 2019, 72, 24-32.

Brown, G.; Reeders, D.; Cogle, A.; Allan, B.; Howard, C.; Rule, J.; Chong, S.; Gleeson, D. Tackling structural stigma: a systems perspective. Journal of the International AIDS Society 2022, 25, e25924.

Brown, G.; Crawford, S.; Perry, G.-E.; Byrne, J.; Dunne, J.; Reeders, D.; Corry, A.; Dicka, J.; Morgan, H.; Jones, S. Achieving meaningful participation of people who use drugs and their peer organizations in a strategic research partnership. Harm reduction journal 2019, 16, 1-10.

Brown, G.; Reeders, D.; Cogle, A.; Madden, A.; Kim, J.; O'Donnell, D. A systems thinking approach to understanding and demonstrating the role of peer-led programs and leadership in the response to HIV and hepatitis C: Findings from the W3 project. Frontiers in Public Health 2018, 6, 231.